# Expression of Inflammatory Genes in Murine Lungs in a Model of Experimental Pulmonary Hypertension: Effects of an Antibody-Based Targeted Delivery of Interleukin-9

**Judith Heiss** [1,2]**, Katja Grün** [1]**, Isabell Singerer** [1]**, Laura Tempel** [1]**, Mattia Matasci** [3]**, Christian Jung** [4]**, Alexander Pfeil** [5]**, P. Christian Schulze** [1]**, Dario Neri** [3] **and Marcus Franz** [1,*]

1   Department of Internal Medicine I, University Hospital Jena, Am Klinikum 1, 07747 Jena, Germany; judith.heiss@uni-jena.de (J.H.); katja.gruen@med.uni-jena.de (K.G.); isabell.singerer@med.uni-jena.de (I.S.); laura.tempel@med.uni-jena.de (L.T.); christian.schulze@med.uni-jena.de (P.C.S.)

2   Else Kröner Graduate School for Medical Students "JSAM", Jena University Hospital, 07747 Jena, Germany

3   Philochem AG, 8112 Otelfingen, Switzerland; mattia.matasci@philogen.com (M.M.)

4   Medical Faculty, Division of Cardiology, Pulmonology and Vascular Medicine, Heinrich-Heine University Düsseldorf, 40225 Düsseldorf, Germany

5   Department of Internal Medicine III, University Hospital Jena, 07747 Jena, Germany; alexander.pfeil@med.uni-jena.de

*   Correspondence: marcus.franz@med.uni-jena.de; Tel.: +49-3641-9324127; Fax: +49-3641-9324142

**Highlights:**

 **What are the main findings?**

- F8IL9 therapy leads to a downregulation of certain inflammatory genes in PH-induced mice and displays beneficial effects on disease severity.

**What is the implication of the main finding?**

- Targeted delivery of IL9 via F8 constitutes a novel treatment strategy in PH which likely has anti-inflammatory and therefore beneficial effects in PH.
- The results of this study offer inspiration to further investigate inflammation and remodeling in PH as a target for future specific therapies.

**Abstract:** Background: Pathogenesis of pulmonary hypertension (PH) is a multifactorial process driven by inflammation and pulmonary vascular remodeling. To target these two aspects of PH, we recently tested a novel treatment: Interleukin-9 (IL9) fused to F8, an antibody that binds to the extra-domain A of fibronectin (EDA⁺ Fn). As EDA⁺ Fn is not found in healthy adult tissue but is expressed during PH, IL9 is delivered specifically to the tissue affected by PH. We found that F8IL9 reduced pulmonary vascular remodeling and attenuated PH compared with sham-treated mice. Purpose: To evaluate possible F8IL9 effects on PH-associated inflammatory processes, we analysed the expression of genes involved in pulmonary immune responses. Methods: We applied the monocrotaline (MCT) model of PH in mice ($n = 44$). Animals were divided into five experimental groups: sham-induced animals without PH (control, $n = 4$), MCT-induced PH without treatment (PH, $n = 8$), dual endothelin receptor antagonist treatment (dual ERA, $n = 8$), F8IL9 treatment ($n = 12$, 2 formats with $n = 6$ each), or with KSFIL9 treatment (KSFIL9, $n = 12$, 2 formats with $n = 6$ each, KSF: control antibody with irrelevant antigen specificity). After 28 days, a RT-PCR gene expression analysis of inflammatory response (84 genes) was performed in the lung. Results: Compared with the controls, 19 genes exhibited relevant (+2.5-fold) upregulation in the PH group without treatment. Gene expression levels in F8IL9-treated lung tissue were reduced compared to the PH group without treatment. This was the case especially for CCL20, CXCL5, C-reactive protein, pentraxin related (CRP$_{PR}$), and Kininogen-1 (KNG1). Conclusion: In accordance with the hypothesis stated above, F8IL9 treatment diminished the upregulation of some genes associated with inflammation in a PH animal model. Therefore, we hypothesize that IL9-based immunocytokine treatment will likely modulate various inflammatory pathways.

**Keywords:** drug delivery; inflammation; gene expression analysis; pulmonary hypertension

## 1. Introduction

Pulmonary hypertension (PH) is defined as an increase of mean pulmonary arterial pressure (mPAP) above 20 mmHg at rest. It can be divided into five clinical groups according to the underlying disease etiology. Specific treatment options only exist for pulmonary arterial hypertension (PAH—group 1 PH) and PH associated with pulmonary artery obstructions (mostly chronic thromboembolic PH, CTEPH—group 4 PH) patients, generally targeting vascular tone to decrease PAP [1]. However, those drugs do not impact PH progression. Moreover, these therapy options cannot be applied to most patients suffering from the very frequent PH groups 2 (associated with left heart disease) and 3 (associated to with disease), leaving them only with the treatment of the underlying disease and supportive care. Consequently, PH patients face poor prognoses as disease progression leads to development of right heart failure. Due to the scarcity of available specific treatment options, there is an unmet clinical need for new therapeutic concepts addressing the pathogenesis of PH [1,2].

Increased vasoconstriction, inflammation, development of thrombosis, and remodeling of pulmonary (artery) vasculature are hallmarks of PH pathogenesis [3,4]. The latter involves a pathological remodeling of all layers of the vessel wall. It is considered an essential factor in development of PH and is described in close relation to inflammatory processes [5,6].

During pulmonary vascular remodeling, a re-expression of fetal matrix proteins, such as the extra domain A (ED-A) containing fibronectin (ED-A$^+$ Fn), can be observed. ED-A$^+$ Fn is a fetal splicing variant of fibronectin which cannot be found in healthy adult tissue. However, it displays a strong re-expression in PH [7,8]. Therefore, it constitutes a promising molecular target to enable, e.g., pharmacodelivery of various payloads selectively to the site of disease using antibodies as vehicle.

In this context, we recently investigated the immunocytokine F8IL9 as a targeted treatment strategy in a mouse model of PH. Interleukin-9 (IL9) hereby constitutes a cytokine which is known to exert pro- and anti-inflammatory effects. IL9 was fused to the F8 antibody specifically recognizing ED-A$^+$ Fn. After the binding of F8 to ED-A$^+$ Fn, IL9 is released due to a change in conformation and can act directly at the site of disease (Figure 1). F8IL9 therapy displayed disease-attenuating effects in comparison to sham-treated controls. This included improvement of hemodynamics and echocardiographic parameters of right heart morphology and function. Pulmonary vascular remodeling was significantly less pronounced after immunocytokine therapy in comparison to untreated PH-induced mice as well [9]. These disease alterations could be observed in context with an accumulation of regulatory T cells in lung tissue of F8IL9-treated mice. This led to the thesis which says that targeted IL9 therapy could attenuate PH-related pulmonary vascular remodeling by mediating anti-inflammatory effects directly at the site of disease in mice [10].

Against the background of F8IL9-associated anti-inflammatory effects, gene expression of inflammatory response and autoimmunity in lung tissue was analysed as an expansion to our previous results.

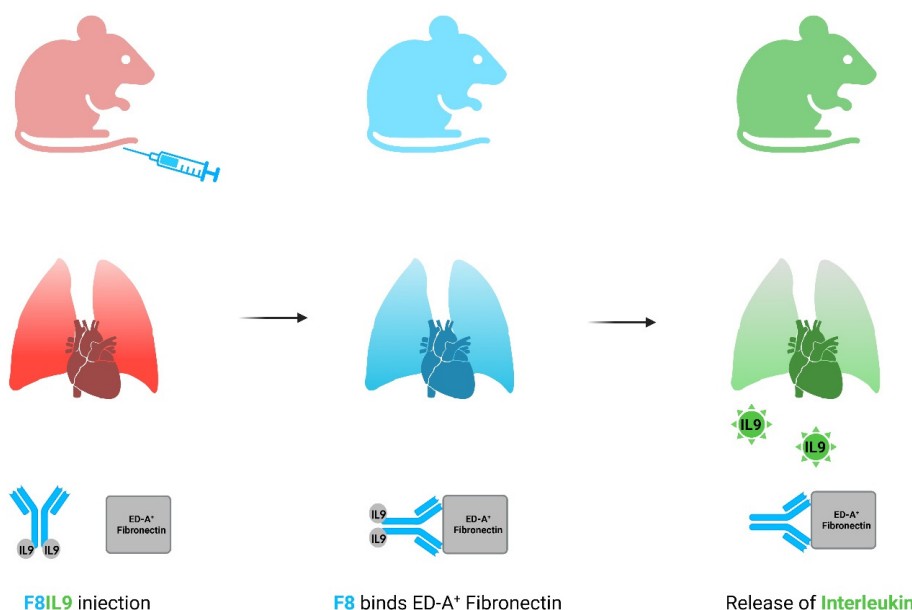

**Figure 1.** Targeted delivery of IL9 via F8-antibody (Figure created by Biorender.com, accessed on 21 October 2023). After F8IL9 injection, F8 bind upon ED-A+ fibronectin expressing tissues such as the lung during PH. Upon binding, a change in the conformation of the immunocytokine leads to a IL9 release. IL9, Interleukin-9.

## 2. Methods

### 2.1. Mouse Model of MCT-Induced PH and Treatment Schedule

We applied the monocrotaline (MCT) model to induce PH in mice ($n = 44$). The animals were divided into the following experimental groups: sham-induced animals without PH (control, $n = 4$), MCT-induced PH without treatment (PH, $n = 8$), dual endothelin receptor antagonist treatment (dual ERA, $n = 8$), F8IL9 treatment ($n = 12$, 2 formats with $n = 6$ each), or KSFIL9 treatment (KSFIL9, $n = 12$, 2 formats with $n = 6$ each). KSF is an antibody with no relevant specificity in mice and served as an isotype control antibody. F8IL9 and KSFIL9 therapy respectively comprise 2 different antibody formats: F8IL9F8 (KSFIL9KSF) with IL9 fused to 2 single chain-variable fragment antibodies and F8IgGIL9 (KSFIgGIL9) with IL9 linked to a full IgG-antibody. Different antibody formats were chosen to evaluate a possible impact of different pharmacokinetic properties on treatment efficacy.

MCT (60 mg/kg body weight, i.p.) was applied as a single shot on day 1 to all PH groups whereas the control group received 30 μL NaCl intraperitoneally. Immunocytokine treatment (F8IL9F8, KSFIL9KSF, F8IgGIL9, KSFIgGIL9) took place on day 14, 16, and 18 (200 μg/injection, intravenously, i.v.). Dual ERA-treated mice received Macitentan (15 mg/kg body weight, per os) from day 14 to 28. On day 28, final in vivo measurements were taken, and the organs were harvested in deep anaesthesia and analgesia. Detailed descriptions of the experimental setup and treatment schedule are given in our previous study [10].

All experiments were conducted according to the National Institute of Health Guidelines for the Care and Use of Laboratory Animals (8th edition), the European Community Council Directive for the Care and Use of Laboratory Animals of 22 September 2010 (2010/63/EU), and the current version of the German Law on the Protection of Animals and the guidelines for animal care. The protocol was approved by the appropriate State Office of Food Safety and Consumer Protection (TLLV, Bad Langensalza, Germany local registration number: UKJ17–003). Moreover, reporting of the study conforms to the EQUATOR guidelines [11].

*2.2. PCR-Based Analysis of Anti-Inflammatory and Autoimmune Genes in Lung Tissue*

We investigated expression of 84 genes of inflammatory response and autoimmunity in lung tissue. After lung tissue maceration, the samples were liquidized with 1 mL TRIzol™ Reagent (Invitrogen; Thermo Fisher Scientific, Inc., Waltham, MA, USA). Acid guanidinium thiocyanate-phenol-chloroform extraction was performed to contain RNA from the tissue [12]. To obtain 500 ng of total RNA, RNA from each experimental group was pooled for cDNA synthesis. RT2 First Strand Kit (Qiagen GmbH, Hilden, Germany) was used according to the manufacturer's instructions to perform cDNA synthesis. Afterwards, cDNA samples were stored at $-20\,^\circ\text{C}$ for further use. Gene expression analysis was performed via $\text{RT}^2$ PCR Array Mouse Inflammatory Response & Autoimmunity (Qiagen GmbH, Hilden, Germany) according to the manufacturer's instructions. For each PCR-Array an Experimental Cocktail (containing 1350 µL 2× $\text{RT}^2$ SYBR-Green ROX Fast Mastermix and 1248 µL RNAse-free water) was added to 102 µL cDNA. 25 µL of this mixture were placed on each well of the $\text{RT}^2$ PCR Array Disc. The disc was then placed in the Cycler Rotor-Gene Q (Qiagen GmbH, Hilden, Germany) and a two-step cycling program was performed including a first cycle of $95\,^\circ\text{C}$ for 10 min followed by 40 cycles of $95\,^\circ\text{C}$ for 15 s and $60\,^\circ\text{C}$ for 60 s. For quality control, a melting curve was generated.

To analyse the data, CT values (cycle threshold value) were calculated for each well ($\text{CT}_{\text{gene}}$) using Software v2.3 (Applied Biosystem/Thermo Fisher Scientific, Darmstadt, Germany). According to the manufacturer's instructions genomic DNA control, reverse transcription control and positive PCR control served as quality control. A mean CT value ($\text{CT}_{\text{HKG}}$) was then generated out of five housekeeping genes (HKG) and used for calculating a $\Delta$CT value ($\Delta\text{CT} = \text{CT}_{\text{gene}} - \text{CT}_{\text{HKG}}$). Heat maps were generated using 40-$\Delta$CT values with matrix2png online interface (available under http://www.chibi.ubc.ca/matrix2png/bin/matrix2png.cgi accessed on 15 March 2022) to visualize data.

To analyse upregulation or downregulation in gene expression in comparison to the control group, $\Delta$CT values of the control group were subtracted from each group ($\Delta\Delta\text{Ct} = \Delta\text{CT}_{\text{PH-induced group}} - \Delta\text{CT}_{\text{control}}$). Then, the fold change was calculated: fold change = $2^{-\Delta\Delta\text{Ct}}$. Fold changes with an absolute value of |2,5| were considered as a relevant change in gene expression compared to the control group.

## 3. Results

Expression of 84 genes of inflammatory response and autoimmunity in lung tissue were analysed. Of these 84 genes, 34 showed a relevant fold change (±2.5-fold) in at least one PH-induced group compared to the control group. Figure 2 summarizes gene expression dynamics in the different groups.

In the untreated PH group, 20 genes exhibited significant changes in expression when compared to the control group. Among these genes, 19 were upregulated. Notably, the most prominent upregulated genes were Chemokine (C-C-motif) ligand 20 (CCL20; +58.9-fold), Chemokine (C-X-C-motif) ligand 5 (CXCL5; +26.1-fold), and Interleukin-17A (IL17A; +7.5-fold) with Chemokine (C-C-motif) ligand 20, Chemokine (C-X-C-motif) ligand 5 (CXCL5; +26.1-fold) and Interleukin-17A (IL17A; +7.5-fold). Only the inducible nitric oxidase synthase 2 (iNOS2) gene displayed a downregulation of expression ($-4.6$-fold).

In general, F8IL9 treatment was linked to lower levels of increased gene expression when compared to the untreated PH group. Twenty-one genes showed less increased gene upregulation in at least one of the two F8IL9-treated groups. This trend of reversibility towards downregulation was best observable for CCL20, CXCL5, C-reactive protein, pentraxin related ($\text{CRP}_{\text{PR}}$), and Kininogen-1 (KNG1). Moreover, $\text{CRP}_{\text{PR}}$ and KNG-1 showed a relevant downregulation of expression in comparison to the control group.

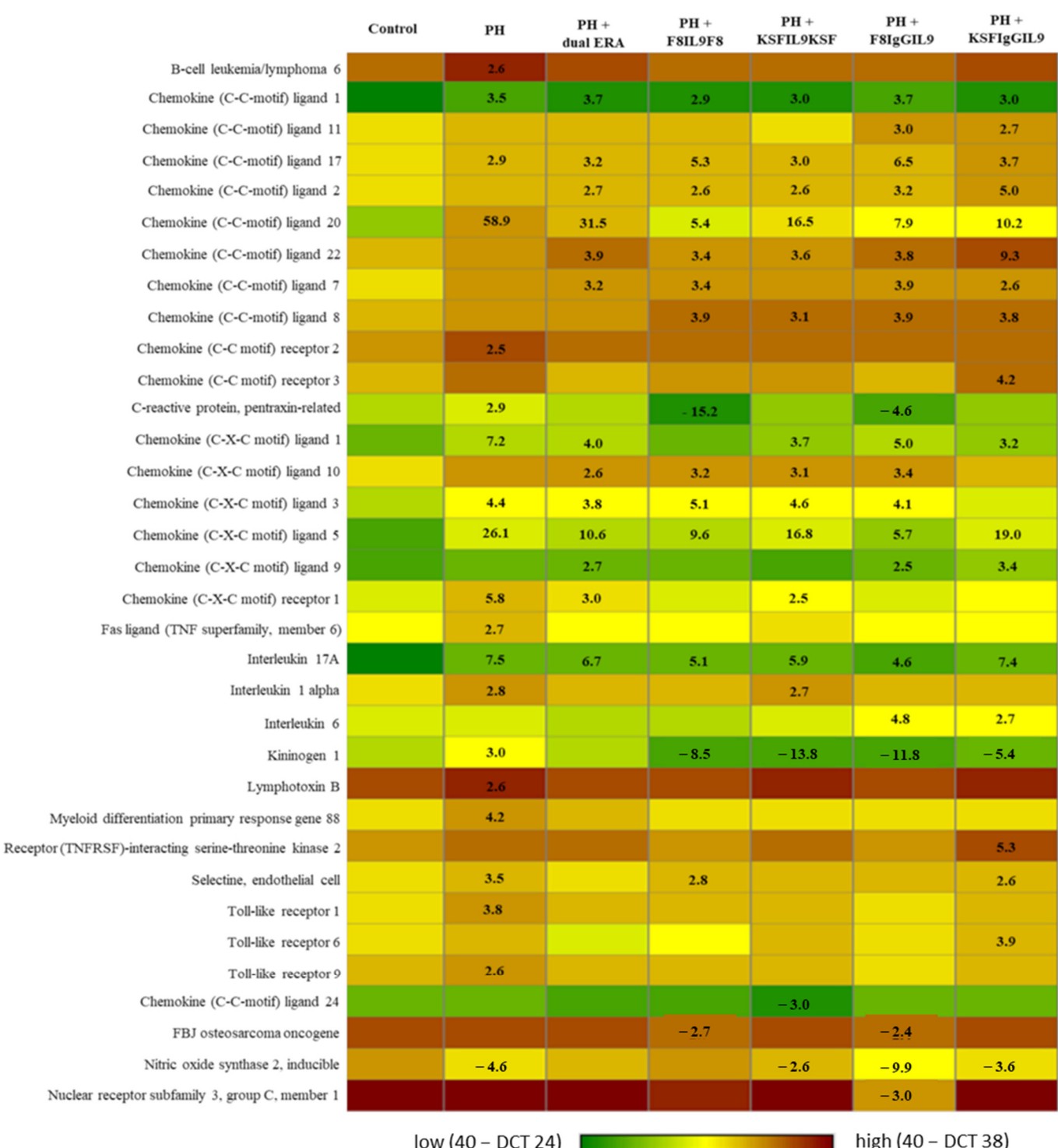

**Figure 2.** Heat-map presentation of relevant gene expression dynamics in the different experimental groups. Fold changes compared to sham-induced controls are given in black. Dual ERA, dual endothelin receptor antagonist; PH, pulmonary hypertension.

Regarding expression in KSFIL9-treated groups, 19 genes showed less increased gene expression levels in at least one of the two KSFIL9 treated PH-induced groups. Though these trends were, in principle, similar to F8IL9-treated mice, it could only be observed to a lesser extent. For instance, fold change of CXCL5 expression in KSFIgGIL9-treated mice (+19-fold) was remarkably lower than in PH-induced group without treatment (+26-fold). Nonetheless, F8IgGIL9- treated mice even demonstrated a lower fold change of CXCL5 expression (+5.9-fold).

In the context of gene expression, the dual ERA group exhibited a reduced fold change when compared with the untreated PH group in 17 genes. However, it's important to note that this effect was less pronounced in comparison to the F8IL9-treated mice.

## 4. Discussion

Our analysis aimed to explore the impact of a specific IL9 transfer on gene expression related to anti-inflammatory responses and autoimmunity in lung tissue. Recently, we hypothesized that F8IL9 mediates its beneficial effects by exerting anti-inflammatory and anti-proliferative properties in preclinical PH. Therefore, this analysis was intended as an expansion to our previous studies [9,10].

The MCT model was chosen to evaluate F8IL9 treatment as it is not only a known PH model, but had already led to significant pulmonary vascular remodeling accompanied by relevant ED-A$^+$ Fn tissue expression in former studies of our group. This ED-A$^+$ Fn deposition made MCT an excellent choice to investigate F8IL9 effects.

We could demonstrate an upregulation of 19 inflammatory genes after PH induction. These results go in line with the current understanding of PH pathogenesis. Thus, inflammation is regarded as one key factor determining the development of PH, and an upregulation of expression of certain inflammatory genes in lung tissue of PH-induced rats has been reported [6,13]. Focusing on single genes, CCL20 was most strikingly upregulated in lung tissue of PH-induced untreated mice. CCL20 is a chemokine which, upon binding with the chemokine receptor 6 (CCR6), recruits inflammatory cells to mucosal sites such as the lung, subsequently leading to T cell activation [14,15]. Perros et al. reported an upregulation in CCL20 mRNA expression and protein levels in lungs of IPAH patients, suggesting an association with the recruitment of IL17-producing cells into the lung [16]. Additionally, a positive correlation of CCL20 serum levels and mPAP in systemic sclerosis patients was observed [17]. Together with our findings, this could identify CCL20 as one mediator of inflammation, driving PH development and presenting a new target for further pathophysiological studies in preclinical PH models. Moreover, CXCL5, and IL17A demonstrated a pronounced upregulation in lung tissue of PH-induced mice without treatment as well. Batah et al. reported significantly elevated IL17 protein levels in lungs shown by immunohistochemistry in MCT-induced rats. In chronic hypoxia models, IL17 inhibition led to an attenuation of PH in mice [18–20]. The IL17A upregulation in lung tissue of PH-induced mice observed in our current analysis is in accordance with these studies. In contrast to IL17, the role of CXCL5 in PH has not been elucidated yet. However, Störmann et al. observed an increase in gene and protein expression after induction of acute lung injuries in mice, which was followed by enhanced neutrophil infiltration. In COPD mice, elevated serum levels of CXCL5 also correlated inversely with lung function [21,22]. Therefore, it seems likely that the chemokine could participate in PH-associated inflammation as well.

Besides PH-induced alterations in mRNA expression, we could show less increased gene expression levels in the lung tissue of F8IL9-treated animals compared to the PH-induced group without treatment. Thus, F8IL9 administration is linked to a reduced upregulation of specific genes related to inflammatory responses and autoimmunity in a preclinical model of PH. We could not observe biologically relevant differences for the majority of genes when comparing F8IL9F8 and F8IgGIL9. This is in line with the results of our previous study. There, we could not demonstrate significant differences between the immunocytokine formats, e.g., concerning echocardiographic parameters or lung tissue damage [10]. Possibly, the different pharmacokinetic properties of the antibodies are not as relevant for F8IL9 treatment outcome in mice or might be too subtle to be depicted with the chosen sample size. However, the differences of these two formats, especially regarding pharmacokinetics, should be subject to further investigations analysing e.g., IL9 plasma concentration, tissue penetration, and half-life in our model.

Compared to F8IL9, similar but less pronounced tendencies in gene expression dynamics could be observed with KSFIL9 treatment. These can be explained with systemic IL9

effects, which have also been observed in our recent study [10]. Mild beneficial effects could be seen for echocardiographic parameters or lung tissue damage after KSFIL9 therapy. Nevertheless, CRP expression was only reduced in F8IL9 treated groups and not in KSFIL9 treated groups when compared to the healthy control. This shows that the targeted delivery of IL9 via F8 is essential for its anti-inflammatory effects.

In mice treated with dual ERA, gene expression exhibited comparable but less pronounced dynamics compared to F8IL9 treatment. The most striking differences between the two groups were observed when examining the gene expression of CCL20, $CRP_{PR}$, and Kininogen. While $CRP_{PR}$ expression is downregulated after F8IL9 treatment compared to the healthy control, its expression is not affected by dual ERA treatment. $CRP_{PR}$ serum levels have been found to be positively correlated with disease severity and adverse outcomes in PH patients [23–25]. In this context, it could be interesting to determine $CRP_{PR}$ serum levels of F8IL9 and dual ERA treated mice to evaluate possible prognostic implications of $CRP_{PR}$ in these therapies. As dual endothelin receptor antagonism mainly addresses vasoconstriction as a pathophysiologic aspect of PH, its effects on inflammatory processes might not be as pronounced as with F8IL9 treatment explaining the less pronounced downregulation of inflammatory gene expression. Since Endothelin-1 has not only been described as a vasoconstrictor but as an inflammatory substance as well, e.g., leading to the recruitment of inflammatory cells into murine lungs [26], dual ERA therapy cannot be reduced due to its impact on vascular tone. It also influences immune homeostasis thus giving rise to the idea of investigating the combination of F8IL9 with dual ERA therapy. Potentially, this could target PH-associated inflammation via two different mechanisms and add anti-vasoconstrictive effects to immunocytokine treatment.

Intriguingly, inducible nitric oxide synthase 2 (iNOS2) expression remained unaffected by dual ERA treatment. However, it exhibited a downregulation within the untreated PH group, as well as in the F8IL9 and KSFIL9 treated groups. The role of nitric oxide (NO) and its synthases in PH remains a topic of contention. On one hand, there exist studies demonstrating that NO-mediated vasodilation exerts beneficial effects. However, on the other hand, evidence suggests that chronic exposure of pulmonary vessels to NO may precipitate vascular remodeling and contribute to disease progression [27]. Since iNOS2 expression is downregulated in untreated PH-induced mice, one could argue that iNOS2 has protective effects in our model. As dual ERA treatment does not show an iNOS2 downregulation, in contrast to F8IL9, combination of dual ERA with F8IL9 again seems a promising approach. Yet one should take caution, as the data is not sufficient to provide mechanistic insights or causality.

In previous studies of F8IL9 effects, we observed an improvement of hemodynamic and echocardiographic parameters and less pronounced pulmonary vascular remodeling after F8IL9 treatment in comparison to PH-induced untreated mice. Moreover, these effects were associated with an accumulation of regulatory T cells (Tregs) in lung tissue. This led us to the assumption that targeted IL9 transfer displays anti-inflammatory properties at the site of disease, which are likely to attenuate pulmonary vascular remodeling during PH and exert beneficial effects on PH progression [10]. Therefore, the presented results of PCR analysis in the lung tissue of F8IL9-treated mice strengthen this thesis. CCL20, CXCL5, $CRP_{PR}$, and Kininogen-1 (KNG1) were the genes most strikingly displaying a less increased upregulation after F8IL9 therapy. $CRP_{PR}$, and KNG-1 even demonstrated a downregulation of gene expression after F8IL9 treatment. Several studies observed that CRP serum levels positively correlate with disease severity and adverse outcomes in PH patients [23–25]. Moreover, it is reported that CRP in human pulmonary arterial smooth muscle cells (PASMCs) induces the production of proinflammatory mediators which are known to contribute to PH development in rat models [28]. Against that background, investigating CRP or one of the other above named inflammatory mediators to a greater depth could contribute to a better understanding of F8IL9-mediated effects in preclinical PH. The findings of this study have to be seen in light of some limitations. First, the experimental set-up does not distinguish F8IL9 effects from possible unspecific effects of

F8 itself. This will be subject of future studies by including a F8 treated group. Secondary, according to unpublished data of our group, ED-A$^+$ Fn expression in secondary organs such as the kidney, liver, or spleen, was only analysed and ruled out in the rat model. Even though there are no reports of ED-A$^+$ Fn being found in other tissue than lung and myocardium during PH, this should be ruled out in mice as well to avoid off-target effects.

**5. Conclusions**

In conclusion, the targeted administration of IL9, which has demonstrated efficacy in treating PH in animal models, likely facilitates a range of beneficial anti-inflammatory mechanisms, ultimately resulting in decreased pulmonary vascular remodeling. Prospectively, gene expression analysis offers inspiration and potential targets for analysis of PH-associated inflammation, therefore leading to a better understanding of MCT-induced alterations in rodents. Moreover, it gives rise to a more thorough analysis of F8IL9-mediated anti-inflammatory effects in the future and hopefully paves the way for novel-disease modifying concepts for the treatment of PH.

**Author Contributions:** J.H., C.J. and M.F. designed the study. J.H., K.G., I.S., L.T. and M.F. performed the experiments and analysed the data. M.M. and D.N. produced and provided immunocytokines. J.H. and M.F. wrote the manuscript. C.J., P.C.S., A.P. and D.N. critically revised the manuscript. All authors have read and agreed to the published version of the manuscript.

**Funding:** This work was supported by funding from the foundation 'Else Kröner-Fresenius-Stiftung' within the Else Kröner Graduate School for Medical Students 'Jena School for Ageing Medicine (JSAM)'.

**Institutional Review Board Statement:** The animal study protocol was approved by the Institutional Review Board of State Office of Food Safety and Consumer Protection (TLLV, Bad Langensalza, Germany local registration number: UKJ17–003, date of approval: 13 September 2017).

**Informed Consent Statement:** Not applicable.

**Data Availability Statement:** The data presented in this study are available in this article or on request from the corresponding author.

**Acknowledgments:** Figure 1 was created with Biorender.com. (accessed on 21 October 2023).

**Conflicts of Interest:** M.M is an employee of Philochem AG (www.philochem.ch), daughter company of Philogen. M.F obtained financial support from Philogen (www.philogen.com). M.F and P.C.S have transferred IP rights to Philogen (www.philogen.com) concerning antibody-cytokine fusions. D.N is CEO and shareholder of Philogen (www.philogen.com), a company working on the discovery and development of targeted therapeutics, including antibody-cytokine fusions.

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
