# Peer review of "Expression of Inflammatory Genes in Murine Lungs in a Model of Experimental Pulmonary Hypertension: Effects of an Antibody-Based Targeted Delivery of Interleukin-9"

_arm, doi:10.3390/arm92010005_

Round 1
Reviewer 1 Report
Comments and Suggestions for Authors
The present study aims to study inflammatory responses after targeting IL9 responses using an ab specific to fibronectin that should, in theory, block IL9 in vascular lesions. The paper is in general well-written and shows the effects of using targeted therapy in an animal model of PH (monocrotaline).
Minor concerns:
- The introduction and discussion should explain why monocrotaline was used and not other models of PH. For example, since IL9 seems to be related to Th2 responses, other animal models, such as Schistosoma-induced PH could be more useful in this context.
- The authors should explain in the text why they used different Ab constructs (F8IL9F8 (KSFIL9KSF) with IL9 fused to 2 single chain-variable fragment antibodies and F8IgGIL9 (KSFIgGIL9) with IL9 linked to a full IgG-antibody)- 2single chain variable fragment abs versus full IgG Ab- do you expect that the variable chain antibodies would be more specific?
- More attention should be given to downstream targets of IL9- that would indicate that the responses were specific for IL9- a flow cytometry experiment showing different T cell responses for example.
- Since they used whole lungs in the RT-PCR, and their target signal is located in the perivascular compartment, the signal may be a little “diluted”
- It would be nice to have confirmation of the results using a protein assay (at least of the markers where the major differences were observed)
Major concerns:
- Many of the markers tested by the RT2 PCR Array Mouse Inflammatory Response & Autoimmunity assay were also decreased in the isotype group in comparison with the PH group (not only in the anti-il9 groups). This implies that there were unspecific effects of the treatment used. Therefore, many of the gene changes in expression resulted from the unspecific effects of the treatment antibodies used. The authors should at least add this as a limitation of the study
- Considering the pleiotropic effects of IL9, I would expect more changes in inflammation-related genes, considering that monocrotaline is a very pro-inflammatory PH model.
Comments on the Quality of English LanguageThe manuscript is nicely written, no major concerns here
Author Response
Dear ladies and gentlemen,
thank you for giving us the opportunity to submit a revised draft of our manuscript titled “Expression of Inflammatory Genes in Murine Lungs in a Model of Experimental Pulmonary Hypertension: Effects of an Antibody-Based Targeted Delivery of Interleukin-9” to “Advances in Respiratory Medicine”.
We appreciate the time and effort that you and the reviewers have dedicated to providing your valuable feedback on our manuscript and are grateful to the reviewers for their insightful comments on our paper. We have been able to incorporate changes to reflect most of the suggestions provided by the reviewers and highlighted these changes within the manuscript (green for changes to avoid similarities, grey for changes suggested by the reviewers). Exact lines of the changes are given in this point-by-point response to the reviewers’ comments and concerns.
______________________________________________________________________________________
Reviewers' Comments to the Authors:
Reviewer 3
Comments and Suggestions for Authors
Authors investigated the MCT model of PH in mice, and compared with gene expression w/o treatment,which is very interesting and can be accepted with minor edition.
Authors response: Thank you!
- Comment: Abstract is too long, authors need to focus more on the important.
Authors response: Thank you for pointing this out. We have shortened the abstract (ll.14-40) and hope that it is now depicting the most important aspects of this study.
- Comment: What's the release mechanism of IL9 from F8IL9.
Authors response: IL9 is released after F8 has bound to ED-A+ Fibronectin due to a change in conformation. You have raised an important point. For the reader’s better understanding, we have therefore included a short explanation in ll. 72-73 and ll. 86-88.
- Comment: What about the systemic toxicity after treating with F8IL9.
Authors response: We agree with the reviewer, that the systemic toxicity is an important aspect for assessing F8IL9 treatment. However, we looked at treatment efficacy of F9IL9 and IL9 downstream effects first, analysis of systemic toxicity will be subject to future investigations.
- Comment: a) Dose ED-A+ Fibronectin overexpression in other tissue? What about the off-target? b) Did author investigate the plasma concentration of IL9 after treating with F8IL9, like the PK parameters?
Authors response: Thank you for these questions.
- ED-A+ Fn can be found during wound healing and angiogenesis as well as a in the female reproductive system. Therefore, only male mice have been investigated According to unpublished data of our group in the rat model, there was no ED-A+ Fn expression in secondary organs, like liver, kidney or spleen. Furthermore, we have found no reports of ED-A+ Fn deposition in other tissue than lung and myocardium during PH. However, this has to be ruled out perspectively in mice, to avoid potential off-target effects. We have included this in the limitations of our study (ll. 288-292).
- We agree with the reviewer that the consideration of PK parameters is of great importance for drug development. Still, as we have explained above, our primary goal was to analyse IL9 treatment efficacy and downstream effects. But we hope to investigate PK parameters in the future (ll. 226-228). In this regard, we think that e.g. Borsi et al have an interesting approach to analyse biodistribution. [5]
- Comment: Details of the experiments treatment and set up need to be given in this manuscript also.
Authors response: We have added information on the experimental setup as suggested (ll. 104-109).
- Comment: After treating with F8IL9, what about the lung function parameters changes, like the echocardiographic parameters, FRC, ERV, et al.
Authors response: This is indeed an interesting idea. Unfortunately, measuring lung function in mice is very challenging and has not performed by us yet. For one, invasive measuring techniques, such as the forced oscillation method, require anaesthesia and expose the animals to a lot of stress. [6] Especially with regard to the final echocardiographic measurements and right heart catheterization, we wanted to avoid that. Sadly, a plethysmograph for non-invasive measurement was not to our disposal. Nonetheless, we primarily focused on analysing echocardiographic parameters of form and function as well as hemodynamic impairment in our recent study to depict right heart function.
_______________________________________________________________________________
References:
[1] Rojas-Zuleta WG, Sanchez E. IL-9: Function, Sources, and Detection: Springer; 2017.
[2] Heiss J, Grün K, Tempel L, Matasci M, Schrepper A, Schwarzer M, et al. Targeted Interleukin-9 delivery in pulmonary hypertension: Comparison of immunocytokine formats and effector cell study. European Journal of Clinical Investigation.n/a:e13907.
[3] Rohm I, Grun K, Muller LM, Baz L, Forster M, Schrepper A, et al. Cellular inflammation in pulmonary hypertension: Detailed analysis of lung and right ventricular tissue, circulating immune cells and effects of a dual endothelin receptor antagonist. Clin Hemorheol Microcirc. 2019;73:497-522.
[4] Nogueira-Ferreira R, Vitorino R, Ferreira R, Henriques-Coelho T. Exploring the monocrotaline animal model for the study of pulmonary arterial hypertension: A network approach. Pulm Pharmacol Ther. 2015;35:8-16.
[5] Castellani P, Borsi L, Carnemolla B, Birò A, Dorcaratto A, Viale GL, et al. Differentiation between high- and low-grade astrocytoma using a human recombinant antibody to the extra domain-B of fibronectin. Am J Pathol. 2002;161:1695-700.
[6] Glaab T, Braun A. Noninvasive Measurement of Pulmonary Function in Experimental Mouse Models of Airway Disease. Lung. 2021;199:255-61.
Reviewer 2 Report
Comments and Suggestions for Authors
Thank you for the oportunity to review this paper.
The aim of this study is to reveal the role of F8IL9 treatment in the atenuation of pulmonary vascular remodelling in the experimental PH.
The results in the F8IL9 treatment group compared to the control group reveal an atenuation of the gene expression levels for CCL 20,CXCL 5 genes and a down-regulation of CRP PR, KNG 1 and NOS2 gene expression. Te comparison with the PH treatment group shows a down regulation of NOS2 gene expression and the ateuation of CCL20,CXCL5,CRP pr and KNG 1 gene expression.
Please extend the discussionabout these results for F8IL9 and F8IgGIL9 treatment groups vs. dual ERA PH group, because of future therapeutic targets of this therapy. Extend also comments about the implications of the attenuation of the CRPpr gene expression and the down-regulation of the NOS2 gene expression compared with the dual ERA treatment group and the KSFIL9 treatment group. Because the multiple pathways PH treatment (by modulating inflamation and pulmonary vascular remodelling) include cocomitent treatment with ERA receptor antagonists and NOS inhibitors, is important to discuss the results of this study in this context, and the possible cummulative effect.
Extend the comments about CRPpr gene expression in the F8IL9 and F8IgGIL9 treatment groups vs ERA treatment group , related to the prognostic implications.
Author Response

(The authors gave the same response as above.)

Reviewer 3 Report
Comments and Suggestions for Authors
Authors investigated the MCT model of PH in mice, and compared with gene expression w/o treatment, which is very interesting and can be accepted with minor edition.
1. Abstract is too long, authors need to focus more on the important.
2. What's the release mechanism of IL9 from F8IL9.
3. What about the systemic toxicity after treating with F8IL9.
4. Dose ED-A+ Fibronectin overexpression in other tissue? What about the off-target? Did author investigate the plasma concentration of IL9 after treating with F8IL9, like the PK parameters?
5. Details of the experiments treatment and set up need to be given in this manuscript also.
6. After treating with F8IL9, what about the lung function parameters changes, like the echocardiographic parameters, FRC, ERV, et al.
Author Response

(The authors gave the same response as above.)

Round 2
Reviewer 1 Report
Comments and Suggestions for Authors
Thanks for the reminder. I agree with the changes made in the manuscript.